# LEARNING UNSUPERVISED FORWARD MODELS FROM OBJECT KEYPOINTS

## ABSTRACT

Object-centric representation is an essential abstraction for forward prediction. Most existing forward models learn this representation through extensive supervision (e.g., object class and bounding box) although such ground-truth information is not readily accessible in reality. To address this, we introduce KINet (Keypoint Interaction Network)—an end-to-end unsupervised framework to reason about object interactions based on a keypoint representation. Using visual observations, our model learns to associate objects with keypoint coordinates and discovers a graph representation of the system as a set of keypoint embeddings and their relations. It then learns an action-conditioned forward model using contrastive estimation to predict future keypoint states. By learning to perform physical reasoning in the keypoint space, our model automatically generalizes to scenarios with a different number of objects, novel backgrounds, and unseen object geometries. Experiments demonstrate the effectiveness of our model in accurately performing forward prediction and learning plannable object-centric representations which can also be used in downstream robotic manipulation tasks.

## 1 INTRODUCTION

Discovering a structured representation of the world allows humans to perform a wide repertoire of motor tasks such as interacting with objects. The core of this process is learning to predict the response of the environment to applying an action (Miall & Wolpert, 1996; Wolpert & Kawato, 1998). The internal models, often referred to as the forward models, come up with an estimation of the future states of the world given its current state and the action. By cascading the predictions of a forward model it is also possible to plan a sequence of actions that would bring the world from an initial state to a desired goal state (Wolpert et al., 1998; 1995).

Recently, deep learning architectures have been proposed to perform forward prediction using an object-centric representation of the system (Ye et al., 2020; Chen et al., 2021b; Li et al., 2020a; Qi et al., 2020). This representation is learned from the visual observation by factorizing the scene into the underlying object instances using ground-truth object states (e.g., object class, position, and bounding box). We identified two major limitations in the existing work: First, they either assume access to the ground-truth object states (Battaglia et al., 2016; Li et al., 2020a) or predict them using idealized techniques such as pre-trained object detection or instance segmentation models (Ye et al., 2020; Qi et al., 2020). However, obtaining ground truth object states is not always feasible in practice. Relying on object detection and segmentation tools, on the other hand, makes the forward model fragile and dependent on the flawless performance of these tools which is often infeasible in real-world settings. Second, factorizing the scene into a fixed number of object instances limits the generalization of the model models to scenarios with a different number of objects.

In this paper, we address both of these limitations by proposing to learn forward models using a keypoint representation. Keypoints represent a set of salient locations of moving entities. Our model KINet (Keypoint Interaction Network) learns an unsupervised forward model in three steps: (1) A keypoint extractor factorizes the scene into keypoints with no supervision other than raw visual observations. (2) A probabilistic graph representation of the system is learned where each node corresponds to a keypoint and edges are keypoints relations. Node features carry implicit object-centric features as well as explicit keypoint state information. (3) With probabilistic message passing, our model learns an action-conditional forward model to predict the future location of keypoints and

reconstruct the future appearances of the scene. We evaluate KINet's forward prediction accuracy and demonstrate that, by learning forward prediction in a keypoint coordinate, our model effectively re-purposes this knowledge and generalizes it to complex unseen circumstances.

Our key contributions are: (1) We introduce KINet, an end-to-end method for learning unsupervised action-conditional forward models from visual observations (2) We introduce probabilistic Interaction Networks for efficient message-passing to aggregate relevant information. (3) We introduce the GraphMPC for accurate action planning using graph similarity. (4) We demonstrate learning forward models in keypoint coordinates enables zero-shot generalization to complex unseen scenarios.

## 2 RELATED WORK

**Unsupervised keypoint extraction.** Keypoints have been used in computer vision areas such as pose tracking (Zhang et al., 2018; Yao et al., 2019) and video prediction (Minderer et al., 2019; Zhang et al., 2018; Xue et al., 2016; Manuelli et al., 2020). Recent work explored keypoints for control tasks in reinforcement learning to project the visual observation to a lower-dimensional keypoint space (Kulkarni et al., 2019; Chen et al., 2021a; Jakab et al., 2018).

**Forward models.** The most fundamentally relevant work to ours is Interaction Networks (IN) (Battaglia et al., 2016; Sanchez-Gonzalez et al., 2018) and follow-up work using graph neural network for forward simulation (Pfaff et al., 2020; Li et al., 2019; Kipf et al., 2018; Mrowca et al., 2018). These methods rely on the ground-truth states of objects to build explicit object representations. Several approaches extended IN by combining explicit object positional information with implicit visual features from images (Watters et al., 2017; Qi et al., 2020; Ye et al., 2020). However, two main concerns remain unaddressed. First, visual features are often extracted from object bounding boxes using object detection or segmentation models (Janner et al., 2018; Qi et al., 2020; Kipf et al., 2018) that are either pretrained on the setup (Qi et al., 2020) or use the ground-truth object position (Ye et al., 2020). Second, these approaches lack generalization as they are formulated on a fixed number of objects. Minderer et al. (2019) used keypoints for video prediction given a history of previous frames. However, their dynamics model is not formulated on external action and cannot be used for action planning applications.

**Action-conditional forward models.** Battaglia et al. (2016) augments the action as an external effect augmented to the node embeddings. Ye et al. (2020) included action as an additional node in a fully connected graph with other nodes representing objects. For probabilistic forward models Henaff et al. (2019) suggests using a latent variable dropout mechanism to properly condition the model on the action (Gal & Ghahramani, 2016). In a more relevant application to ours, Yan et al. (2020) highlighted the effectiveness of contrastive estimation (Oord et al., 2018) to learn proper actionable object-centric representations.

**Unsupervised Forwad Models.** Kipf et al. (2019) uses an object-level contrastive loss to learn object-centric abstractions in multibody environments with deterministic structures and minimal visual features such as 2d shapes. In our work, we randomize the properties of the system and examine realistic objects. Veerapaneni et al. (2020) infers a set of entities in the image based on their depth and predicts future entity states. Entities are mixed using weight parameters of their distance from the camera. Our work, on the other hand, does not make any assumption on inferring depth and relies on keypoints. Kossen et al. (2019) uses images to infer a set of explicit states (e.g., position and velocity) for a fixed number of objects in a dynamic system to predict the future state of each object using graph networks. Although learning unsupervised state representation, this method is formulated on a fixed number of objects and only tested on environments with simple 2d geometries. Li et al. (2020b) infers the causal structure and makes future predictions on a fixed dynamic system using a pretrained keypoint extractor on topview images. In our work, we experiment with other camera angles and 3d objects with random properties such as geometry and texture.

## 3 KEYPOINT INTERACTION NETWORKS (KINET)

We assume access to observational data that consists of RGB image, action vector, and the resulting image after applying the action: $\mathcal{D} = \{(I_t, u_t, I_{t+1})\}$. Our goal is to learn a forward model that predicts the future states of the system with no supervision above the observational data. We describe our approach in two main steps (see Figure 1): learning to encode visual observations into keypoints and learning an action-conditioned forward model in the keypoint space.

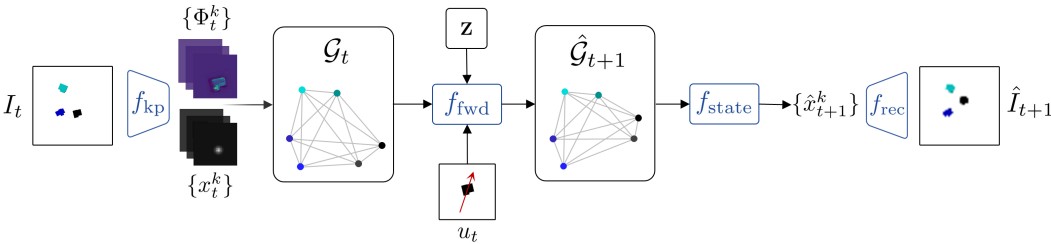

Figure 1: KINet performs forward modeling in three major steps: extracting keypoints, inferring a probabilistic graph representation, and estimating the future appearance conditioned on the action. Some arrows are simplified for clarity.

### 3.1 UNSUPERVISED KEYPOINT DETECTION

The keypoint detector ($f_{\text{kp}}$, Fig.1) is a mapping from visual observations to a lower-dimensional set of $K$ keypoint coordinates $\{x_t^k\}_{k=1\dots K} = f_{\text{kp}}(I_t)$. The keypoint coordinates are learned by capturing the spatial appearance of the objects in an unsupervised manner. Specifically, the keypoint detector receives a pair of initial and current images $(I_0, I_t)$ and uses a convolutional encoder to compute a $K$-dimensional feature map for each image $\Phi(I_0), \Phi(I_t) \in \mathbb{R}^{H' \times W' \times K}$. The expected number of keypoints is set by the dimension $K$. Next, the feature maps are marginalized into a 2D keypoint coordinate $\{x_0^k, x_t^k\}_{k=1\dots K} \in \mathbb{R}^2$. We use a convolutional image reconstruction model ($f_{\text{rec}}$, Fig.1) with skip connections to inpaint the current image frame using the initial image and the predicted keypoint coordinates $\hat{I}_t = f_{\text{rec}}(I_0, \{x_0^k, x_t^k\}_{k=1\dots K})$. With this formulation, $f_{\text{kp}}$ and $f_{\text{rec}}$ create a bottleneck to encode the visual observation in a temporally consistent lower-dimensional set of keypoint coordinates (Kulkarni et al., 2019).

### 3.2 GRAPH REPRESENTATION OF SYSTEM

After factorizing the system into $K$ keypoints, we build a graph $\mathcal{G}_t = (\mathcal{V}_t, \mathcal{E}_t, \mathbf{Z})$ (undirected, no self-loop) where keypoints and their pairwise relations are the graph nodes and edges. Keypoint positional and visual information are encoded into embedding of nodes $\{\mathbf{n}_t^k\}_{k=1\dots K} \in \mathcal{V}_t$ and edges $\{\mathbf{e}_t^{ij}\} \in \mathcal{E}_t$. We also use an adjacency matrix to specify the connectivity as $\mathbf{Z} \in \mathbb{R}^{K \times K}$ where $[\mathbf{Z}]_{ij} \in [0, 1]$ specifies the probability of the edge $\{\mathbf{e}^{ij}\} \in \mathcal{E}$. At timestep $t$, node embeddings encode keypoint visual and positional information $\{\mathbf{n}_t^k\} = [\Phi_t^k, x_t^k]$. Edge embeddings contain relative positional information of each node pair $\{\mathbf{e}_t^{ij}\} = [x_t^i - x_t^j, \|x_t^i - x_t^j\|_2^2]$.

### 3.3 PROBABILISTIC INTERACTION NETWORKS

To build a forward model, we extend the recent approaches and propose a probabilistic variation of the Interaction Networks (IN) (Battaglia et al., 2016; Sanchez-Gonzalez et al., 2018). The core of probabilistic IN is generating edge-level latent variables $z_{ij} \in \mathbb{R}^d$ that represents the edge probability $[\mathbf{Z}]_{ij} = z_{ij}$. A posterior network $p_\theta$ infers the elements of the adjacency matrix given the graph representation of the scene. In particular, we model the posterior as $p_\theta(z_{ij}|\mathcal{G}_t) = \sigma(f_{\text{enc}}([\mathbf{n}_t^i, \mathbf{n}_t^j]))$ where $\sigma(.)$ is the sigmoid function.

The probabilistic IN forward model $\hat{\mathcal{G}}_{t+1} = f_{\text{fwd}}(\mathcal{G}_t, u_t, \mathbf{Z})$ predicts the graph representation at the next timestep by taking as input the current graph representation and action ($f_{\text{fwd}}$, Fig.1). The message-passing operation in the forward model can be described as, $\{\hat{\mathbf{e}}^{ij}\} \leftarrow f_{\text{e}}(\mathbf{n}^i, \mathbf{n}^j, \mathbf{e}^{ij})$, $\{\hat{\mathbf{n}}^k\} \leftarrow f_{\text{n}}(\mathbf{n}^k, \sum_{i \in N(k)} \mathbf{Z}_{ik}\hat{\mathbf{e}}^{ik}, u_t)$ where the edge-specific function $f_{\text{e}}$ first updates edge embeddings, then the node-specific function $f_{\text{n}}$ updates each node embedding $\hat{\mathbf{n}}^k$ by probabilistically aggregating its neighboring nodes $N(k)$ information (i.e., the edge probabilities from the inferred adjacency matrix $\mathbf{Z}$ is used as weights in the neighbor aggregation). The action vector $u_t$ is also an input to the neighbor aggregation step. Note that the functions $f_{\text{enc}}$, $f_{\text{e}}$, and $f_{\text{n}}$ are MLPs.

Recent models for forward prediction rely on fully connected graphs for message passing (Qi et al., 2020; Ye et al., 2020; Li et al., 2018). Our model, however, learns to probabilistically sample the

neighbor information. Intuitively, this adaptive sampling allows the network to efficiently aggregate the most relevant neighboring information. This is specifically essential in our model since keypoints could provide redundant information if they are in very close proximity.

### 3.4 FORWARD PREDICTION

The state decoder ($f_{\text{state}}$, Fig.1) transforms the predicted node embeddings of the updated graph $\hat{\mathcal{G}}_{t+1}$ to a first-order difference which is integrated once to predict the position of the keypoints in the next timestep $\{\hat{x}_{t+1}^k\} = \{x_t^k\} + f_{\text{state}}(\{\hat{\mathbf{n}}_{t+1}^k\})$. To reconstruct the image at the next timestep, we borrow the reconstruction model $f_{\text{rec}}$ from the keypoint detector $\hat{I}_{t+1} = f_{\text{rec}}(I_0, \{x_0^k, \hat{x}_{t+1}^k\})$.

### 3.5 LEARNING KINET

**Reconstruction loss.** The keypoint detector is trained using the distance between the ground truth image and the reconstructed image at each timestep $\mathcal{L}_{\text{rec}} = \|\hat{I}_t - I_t\|_2^2$. As suggested by Minderer et al. (2019), errors from the keypoint detector were not backpropagated to other modules of the model. This is a necessary step to ensure the model does not conflate errors from image modules and reasoning modules.
**Forward loss.** The model is also optimized to predict the next state of the keypoints. A forward loss penalizes the distance between the estimated future keypoint locations using first-order state decoder and the keypoint extractor predictions: $\mathcal{L}_{\text{fwd}} = \sum_K \|\hat{x}_{t+1}^k - f_{\text{kp}}(I_{t+1})^k\|_2^2$.
**Inference loss.** Our model is also trained to minimize the KL-divergence between the posterior and prior distributions: $\mathcal{L}_{\text{infer}} = D_{\text{KL}}\big(p_\phi(\mathbf{Z}|\mathcal{G})\big\|p(\mathbf{Z})\big)$. We use independent Gaussian prior $p(\mathbf{Z}) = \prod_i \mathcal{N}(\mathbf{z}_i)$ and use reparameterization trick for training (Kingma & Welling, 2013).

**Contrastive loss.** We use the contrastive estimation method to further enhance the learned graph representations. We add a contrastive loss (Oord et al., 2018; Yan et al., 2020) and reframe it for graph embeddings as: $\mathcal{L}_{\text{ctr}} = -\mathbb{E}_\mathcal{D}[\log(\mathcal{S}(\hat{\mathcal{G}}_{t+1}, \mathcal{G}_{t+1}^+)/\sum \mathcal{S}(\hat{\mathcal{G}}_{t+1}, \mathcal{G}_{t+1}^-))]$. The predicted graph representations $\hat{\mathcal{G}}_{t+1}$ are maximally similar to their corresponding positive sample pair $\mathcal{G}_{t+1}^+ := \mathcal{G}_{t+1}$ and maximally distant from the negative sample pairs $\mathcal{G}_{t+1}^- := G_\tau \ \forall \tau \neq t+1$. We use a simple node embedding similarity as the graph matching algorithm $\mathcal{S}(\mathcal{G}_1, \mathcal{G}_2) = \sum_K \{n_1^k\} \cdot \{n_2^k\}$. The motivation behind adding a contrastive loss is aligning the graph representation of similar object configurations while pushing apart those of dissimilar configurations in the embedding space to enhance the learned graphs.

Finally, the combined loss is: $\mathcal{L} = \lambda_{\text{rec}} \mathcal{L}_{\text{rec}} + \lambda_{\text{fwd}} \mathcal{L}_{\text{fwd}} + \lambda_{\text{infer}} \mathcal{L}_{\text{infer}} + \lambda_{\text{ctr}} \mathcal{L}_{\text{ctr}}$.

### 3.6 GRAPHMPC PLANNING WITH KINET

We use a learned KINet model and select actions based on Model Predictive Control (MPC) algorithm (Finn & Levine, 2017) in the graph embedding space (GraphMPC). With KINet forward prediction, we compute the predicted graph representation of the next timestep for multiple sampled actions. We then select the optimal action that produces the most similar graph representation to the goal graph representation $\mathcal{G}^{\text{goal}}$. We describe our GraphMPC algorithm with a time horizon of $T$ as: $u_t^* = \arg\max\{\mathcal{S}\big(\mathcal{G}^{\text{goal}}, f_{\text{fwd}}(\mathcal{G}^t, \{u_{t:T}\})\big)\}; \ t \in [0, T]$. Unlike performing conventional MPC only with respect to positional states, GraphMPC allows for accurately bringing the system to a goal state both explicitly (i.e, position) and implicitly (i.e, pose, orientation, and visual appearance).

## 4 EXPERIMENTAL SETUPS

Our experiments seek to address the following: (1) How accurate is our forward model? (2) Can we use the model for action planning? and (3) Does the model generalize to unseen circumstances?

### 4.1 DATASET

We apply our approach to learn a forward model for multi-object manipulation tasks. The task involves rearranging multiple objects in the scene and bringing them to the desired goal state using

pushing actions. In the *RealBlocks* dataset, we use the Sawyer robot pushing dataset in Ye et al. (2020) to exemplify how our proposed framework applies to real settings. *RealBlocks* data includes 5K random pushing actions on 7 blocks. For each action, an RGB image pair with the action vector in the image coordinates is captured.

We use MuJoCo 2.0 (Todorov et al., 2012) to generate two sets of simulated scenarios. In *BlockObjs* dataset, we simulate 10K episodes of random pushing on multiple objects (1-5 objects) where a simple robot end-effector applies randomized pushing for 90 timesteps per episode (Figure A.1.2). Each object has randomized geometry and color. We sample the object geometry from a predefined continuous range denoted as $\text{geom}_{train}$ for training and $\text{geom}_{gen}$ for generalization to unseen geometries. Unseen geometries are designed to have elongated shapes to create complex out-of-distribution cases (Figure A.1.1). In *YCBObjs* dataset, we simulate 10K episodes (60 timesteps) of randomly pushing a subset of YCB objects placed on a wooden table (1-5 objects) (Calli et al., 2015) that includes objects of daily life with diverse properties such as shape, size, and texture (Figure A.1.3). We collect the 4-dimensional action vector (pushing start and end location) and RGB images before and after each action is applied. Images are obtained using an overhead (*Top View*) and an angled camera (*Angled View*) (see Appendix A.1 for more details).

## 4.2 BASELINES

We compare our approach with existing methods for learning object-centric forward models:

**Forward Model** *(Forw)*: We train a convolutional encoder to extract visual features of the scene image (*Img*) and learn a forward model in the feature space.
**Forward-Inverse Model** *(ForwInv)*: We train a convolutional encoder to extract visual features of the scene image (*Img*) and jointly learn forward and inverse models Agrawal et al. (2016).
**Interaction Network** *(IN)*: We follow Battaglia et al. (2016); Sanchez-Gonzalez et al. (2018) to build an Interaction Network based on the ground truth location of the objects. Each object representation contains the ground-truth position and velocity of the objects (*GT state*). This approach is only applicable to scenarios where the number of objects in the scene is known and fixed.
**Visual Interaction Network** *(VisIN)*: To replicate Ye et al. (2020); Watters et al. (2017), we train a convolutional encoder to extract visual features of fixed-size bounding boxes centered on ground-truth object locations (*GT state + Img*). We use the extracted visual features as object representations in the Interaction Network. This approach requires prior knowledge of the number of objects.
**Causal Discovery from Videos** *(V-CDN)*: Li et al. (2020b) infers the causal structure of a fixed physical system from visual observations and makes future predictions for that system. A pretrained perception module extracts keypoints that are used in an inference module to predict a graph representation for the visual observation which is then used in an IN-based dynamics module to predict the future location of the keypoints.

## 4.3 TRAINING AND EVALUATION SETTING.

All models are trained on a subset of the simulated data (*BlockObjs* and *YCBObjs*) with 3 objects (8K episodes: 80% training, 10% validation, and 10% testing sets). To evaluate generalization to a different number of objects, we use other subsets of data with 1, 2, 4, and 5 objects ($\sim 400$ episodes for each case). We train our model separately on images obtained from the overhead camera (*Top View*) and the angled camera (*Angled View*). For *RealBlocks* data, we only provide qualitative results as the ground-truth location of the objects is unknown. We set the expected number of keypoints $K = 6$ for *BlockObjs*, $K = 9$ for *YCBObjs*, and $K = 14$ for *RealBlocks*.

## 5 RESULTS

This section is organized to answer a series of questions to thoroughly evaluate our model and justify the choices we made in formulating our approach.

### 5.1 DOES THE MODEL ACCURATELY LEARN A FORWARD MODEL?

First, we evaluate the forward prediction accuracy. Figure 2 showcases qualitative results of forward predictions on *RealBlocks*. Our model factorizes the observation into a keypoint representation

and accurately estimates the future appearance of the scene conditioned on external action. The qualitative results highlight that our model learns the effect of the action on objects as well as object-to-object interactions (see Appendix A.4 for more examples).

$$I_t \qquad \{x_t^k\} \qquad u_t, \{\hat{x}_{t+1}^k\} \qquad \hat{I}_{t+1} \qquad I_{t+1}$$

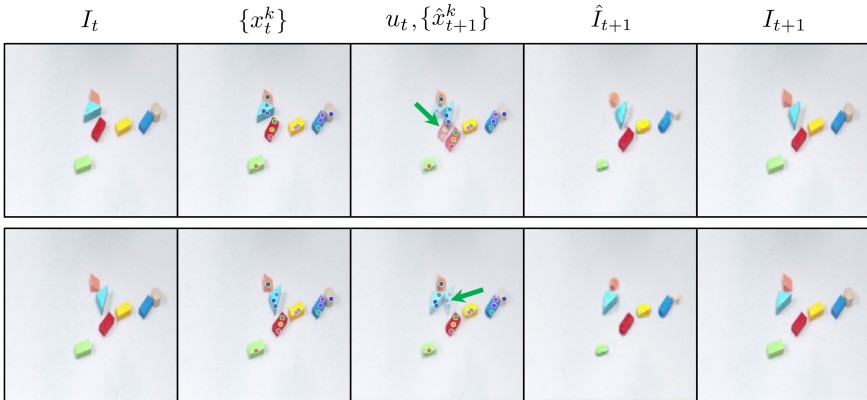

Figure 2: Qualitative results on *RealBlocks* dataset (Ye et al., 2020). Given an image $I_t$, our model factorizes the scene into keypoints $x_t^k$ and conditioned on the action $u_t$ (green arrows) estimates the next keypoint coordinates $\hat{x}_{t+1}^k$ and appearance $\hat{I}_{t+1}$ of the scene.

Table 1: Forward Prediction performance on *BlockObjs* measured as single-step predictions error.

| Model | Supervision | Mean Position Error $\times 10^{-3}$ | |
| --- | --- | --- | --- |
| | | *Top View* | *Angled View* |
| Forw | Img | $0.309_{\pm 0.12}$ | $0.317_{\pm 0.09}$ |
| ForwInv | Img | $0.293_{\pm 0.08}$ | $0.266_{\pm 0.02}$ |
| IN | GT state | $0.112_{\pm 0.003}$ | $0.109_{\pm 0.008}$ |
| VisIN | GT state + Img | $0.107_{\pm 0.006}$ | $0.121_{\pm 0.03}$ |
| KINet (Ours) | Img | $0.122_{\pm 0.01}$ | $0.129_{\pm 0.02}$ |
| KINet - deterministic | Img | $0.127_{\pm 0.03}$ | $0.133_{\pm 0.01}$ |
| KINet - no ctr loss | Img | $0.173_{\pm 0.02}$ | $0.169_{\pm 0.05}$ |

We quantify the effectiveness of our model in comparison with Forw, ForwInv, IN, and VisIN baselines (Table 1) on *BlockObjs* data. We separately train and examine each model on *Top View* and *Angled View* images. The prediction error is computed as the average distance between the predicted and ground-truth positional states. VisIN performs the best among baseline models as it builds object representations with explicit ground-truth object positions and their visual features. Our model, on the other hand, achieves a comparable performance to VisIN while it does not rely on any supervision beyond the scene images. Forw and ForwInv baselines have similar supervision to ours but are significantly less accurate. This emphasizes the capability of our approach in learning a rich graph representation of the scene and an accurate forward model while relaxing prevailing assumptions of the prior work on the structure of the environment and availability of ground-truth state information (see Appendix A.2 for the learned graph representations).

## 5.2 CAN WE USE THE MODEL IN CONTROL TASKS?

We design a robotic manipulation task of rearranging a set of objects to the desired goal state using MPC with pushing actions. For all models, we run 1K episodes with randomized object geometries and initial poses and a random goal configuration of objects. The planning horizon is set to $T = 40$ timesteps in each episode. For our model, we perform GraphMPC based on graph embedding similarity as described in Section 3.6. For all baseline models, we perform MPC directly on the distance to the goal. Figure 5.2-a shows MPC results of *BlockObjs* based on *Top View* observations. Our approach is consistently faster than the baseline models in reaching the goal configuration.

## 5.3 DOES THE MODEL GENERALIZE TO UNSEEN CIRCUMSTANCES?

One of our main motivations to learn a forward model in keypoint space is to eliminate the dependency of the model formulation on the number of objects in the system which allows for generalization to an unseen number of objects, object geometries, background textures, etc.

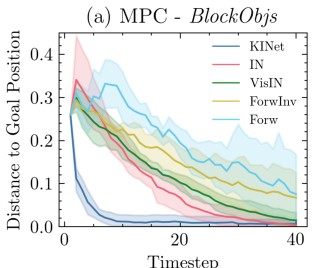 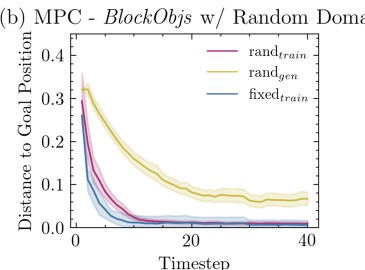

Figure 3: MPC results on *BlockObjs* measured as the distance to goal configuration. (a) Comparison with baselines. (b) MPC for KINet trained on a fixed white background ($\text{fixed}_{train}$), generalization to random backgrounds ($\text{rand}_{gen}$), and trained on random backgrounds ($\text{rand}_{train}$).

***BlockObjs***. We train KINet on 3 randomized blocks with ($\text{geom}_{train}$) and test for zero-shot generalization to an unseen number of objects (1, 2, 4, 5), unseen object geometries ($\text{geom}_{gen}$), and unseen background texture. Figure 4 shows qualitative generalization results. We separately train and examine for generalization on *Top View* and *Angled View* observations with a planning horizon of $T = 80$. Since our model learns to perform forward modeling in the keypoint space, it reassigns the keypoints to unseen objects and makes forward predictions. Table 2 summarizes the generalization performance. As expected, by increasing the number of objects the average distance to the goal position increases. Also, objects with out-of-distribution geometries have more distance to the goal position. Our model's generalization performance is significantly superior to Forw baseline which is a simple action-conditioned video prediction model that does not depend on the number of objects and uses the same supervision as our model.

Further, we test the performance of the model on unseen background textures (i.e., the table texture). Since the keypoint extraction relies on visual features of salient objects, our model is able to perform the control tasks by ignoring the background and assigning keypoints to the moving objects. Figure 5.2-b compares the MPC results for the KINet trained on a fixed white background ($\text{fixed}_{train}$), zero-shot generalization of KINet trained on the fixed background to randomized backgrounds ($\text{rand}_{gen}$), and KINet trained directly on randomized backgrounds ($\text{rand}_{train}$). As expected, although the MPC converges, the final distance to the goal configuration is larger for $\text{rand}_{gen}$. This final error is statistically at the same level of accuracy for $\text{fixed}_{train}$ and $\text{rand}_{train}$. For qualitative examples, see Figure 4 (third row) and Figure A.5.2.

Table 2: Generalization results measured as the average distance to the goal position.

| | | KINet (Ours) | | KINet - deterministic | | Forw | |
|---|---|---|---|---|---|---|---|
| | Objects | $\text{geom}_{train}$ | $\text{geom}_{gen}$ | $\text{geom}_{train}$ | $\text{geom}_{gen}$ | $\text{geom}_{train}$ | $\text{geom}_{gen}$ |
| *Top View* | 1 | $0.24_{\pm0.02}$ | $0.31_{\pm0.01}$ | $0.25_{\pm0.02}$ | $0.34_{\pm0.05}$ | $1.18_{\pm0.12}$ | $1.13_{\pm0.07}$ |
| | 2 | $0.22_{\pm0.01}$ | $0.58_{\pm0.02}$ | $0.26_{\pm0.02}$ | $0.65_{\pm0.03}$ | $1.07_{\pm0.08}$ | $1.02_{\pm0.05}$ |
| | 3 | $0.18_{\pm0.03}$ | $0.21_{\pm0.01}$ | $0.19_{\pm0.06}$ | $0.28_{\pm0.08}$ | $0.65_{\pm0.13}$ | $0.89_{\pm0.11}$ |
| | 4 | $0.54_{\pm0.01}$ | $0.63_{\pm0.13}$ | $0.68_{\pm0.09}$ | $0.89_{\pm0.11}$ | $1.87_{\pm0.15}$ | $1.96_{\pm0.13}$ |
| | 5 | $0.86_{\pm0.08}$ | $1.73_{\pm0.16}$ | $0.94_{\pm0.06}$ | $2.01_{\pm0.14}$ | $1.95_{\pm0.12}$ | $3.12_{\pm0.10}$ |
| *Angled View* | 1 | $0.21_{\pm0.04}$ | $0.35_{\pm0.04}$ | $0.28_{\pm0.01}$ | $0.36_{\pm0.08}$ | $1.09_{\pm0.08}$ | $1.34_{\pm0.07}$ |
| | 2 | $0.21_{\pm0.03}$ | $0.53_{\pm0.06}$ | $0.22_{\pm0.05}$ | $0.59_{\pm0.07}$ | $1.12_{\pm0.12}$ | $1.27_{\pm0.09}$ |
| | 3 | $0.19_{\pm0.02}$ | $0.20_{\pm0.05}$ | $0.19_{\pm0.03}$ | $0.31_{\pm0.08}$ | $0.98_{\pm0.11}$ | $1.16_{\pm0.09}$ |
| | 4 | $0.51_{\pm0.02}$ | $0.65_{\pm0.07}$ | $0.57_{\pm0.12}$ | $0.96_{\pm0.09}$ | $1.33_{\pm0.15}$ | $2.25_{\pm0.13}$ |
| | 5 | $0.89_{\pm0.13}$ | $1.64_{\pm0.11}$ | $1.05_{\pm0.16}$ | $2.17_{\pm0.10}$ | $1.48_{\pm0.12}$ | $3.20_{\pm0.19}$ |

Distance to Goal Position $\times 10^{-3}$

***YCBObjs***. We evaluate the generalization of our model on a set of realistic YCB objects (Calli et al., 2015) with challenging diverse properties such as color, texture, mass, and geometry. We train our model on a random subset of 3 YCB objects and test for generalization to an unseen number of objects (1,2,4,5) with a planning horizon of $T = 40$. As shown in Figure 5, our method generalizes well to an unseen number of objects and performs the control task accurately. Importantly, assigning multiple keypoints to each object allows our framework to implicitly capture the orientation of each object, as well as their position without any supervision on the object pose (e.g., compare the power drill pose in Fig, 5).

We compare the performance of our framework with V-CDN (Li et al., 2020b) baseline which is also a keypoint-based model to learn the structure of physical systems and perform future predictions and

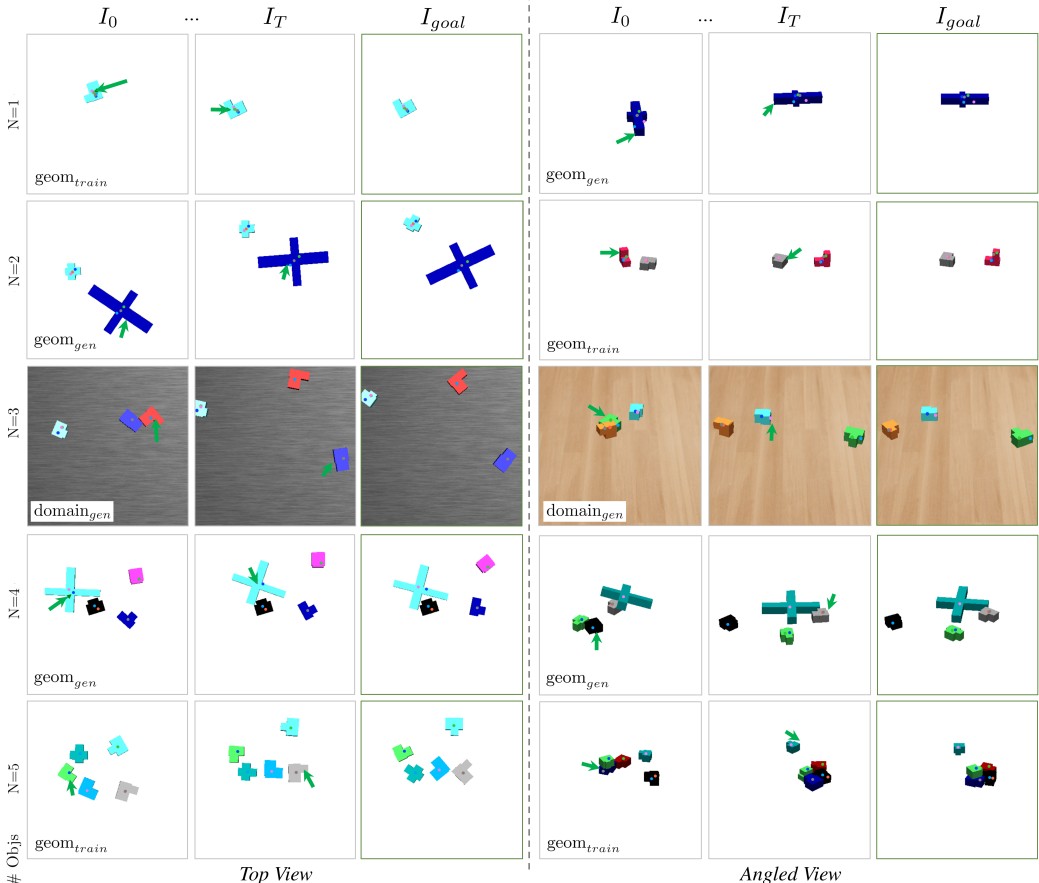

Figure 4: Qualitative results of generalization on *BlockObjs* to unseen number of objects, geometries (geom$_{gen}$) and background textures (domain$_{gen}$). The green arrows are the optimal actions.

potentially generalize to an unseen number of objects. Although V-CDN is formulated to extract the causal structure of a fixed system through visual observations, we pretrain its perception module on our randomized *YCBObjs* dataset for a direct comparison to our model. Also, to perform a control task, we condition the V-CDN on the action by adding an encoding of the action vector to each keypoint embedding. Figure 6 compares the MPC results (normalized to the number of objects) of our method and V-CDN both trained on a subset of 3 YCB objects. The MPC performance of both models is comparable for rearranging 3 objects (green lines). However, V-CDN is most accurate for the number of objects it has been trained on and significantly less accurate when generalizing to an unseen number of objects. We attribute this to two reasons: (1) Although using keypoints, V-CDN is formulated to infer explicit causal structure and attributes for the graph representation of a fixed system setup which does not necessarily carry over to unseen circumstances. (2) Unlike our method, V-CDN does not take into account the visual features in the model and only uses the keypoint positions.

## 5.4  ANALYSIS AND ABLATION

We justify the major choices we made to formulate the model with ablation studies. We examine two elements in our approach: the probabilistic graph representation, and the contrastive loss. We train two variants of KINet: (1) *KINet - deterministic* with a fully connected graph instead of probabilistic (2) *KINet - no ctr loss* without the contrastive loss. The best forward prediction for both *Top View* and *Angled View* images is achieved when the model is probabilistic and trained with a contrastive loss (Table 1). The contrastive loss is an essential element in our approach to ensure the learned forward model is accurately action conditional. Also, with a probabilistic graph representation, our model achieves better generalization compared to the deterministic variant. This performance gap is more evident when generalizing to unseen geometries (Table 2).

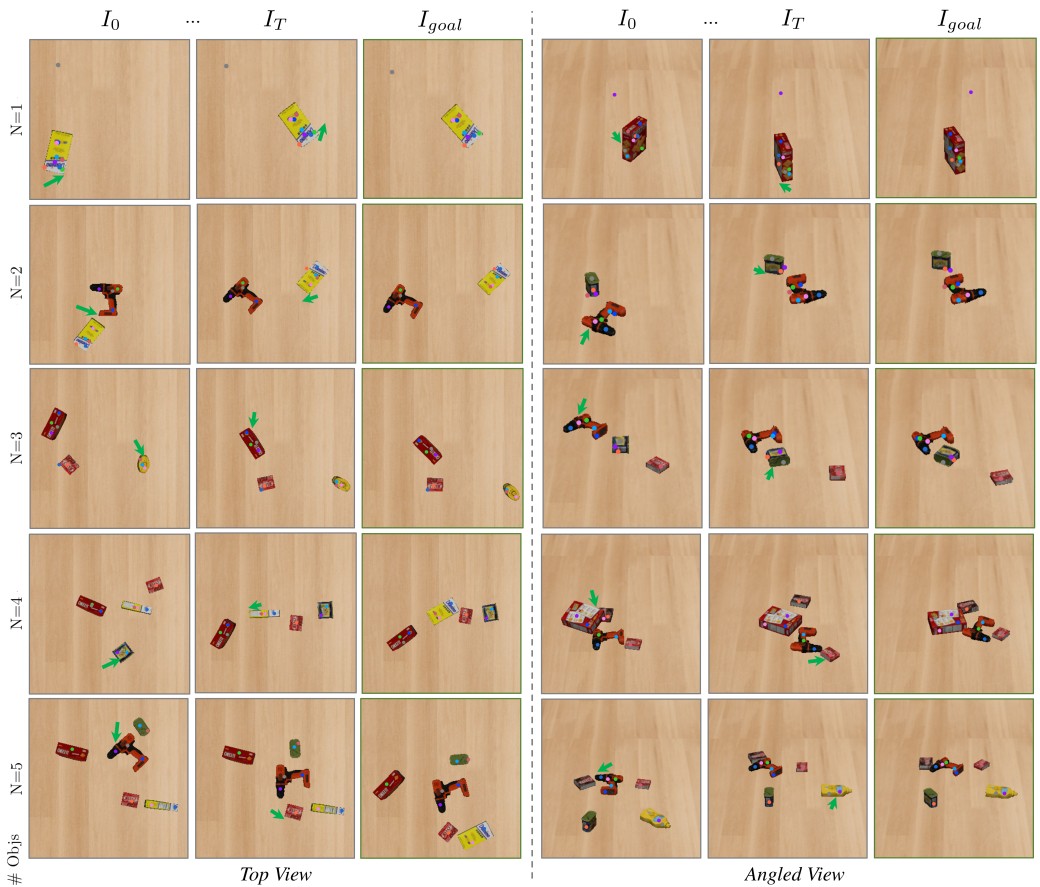

Figure 5: Qualitative results of generalization on *YCBObjs* to unseen number of objects (1,2,4,5) for *Top* and *Angled View* observations.

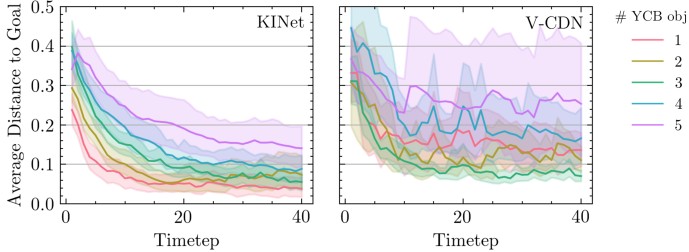

Figure 6: *YCBObjs* MPC results for generalization to a varying number of objects.

# 6 CONCLUSION

In this paper, we proposed a method for learning action-conditioned forward models based only on image observations. We showed that our approach effectively makes forward predictions through keypoint factorization. Also, we demonstrated that a keypoint-based forward model, unlike prior work, does not make assumptions about the number of objects which allows for automatic generalization to a variety of unseen circumstances. Importantly, our model learns a forward model without explicit supervision on ground-truth object state information. One limitation in our formulation is fixing the number of expected keypoints (see Appendix A.3). However, we showed this gives more generalizability compared to fixing the number of objects. We also observed inconsistency in the predicted keypoints for real-robot data scenarios where most of the objects were pushed out of the image frame (see Appendix A.6). An interesting future direction is to focus on the keypoint extraction module to further enhance forward models for real settings. Finally, we hope our general approach inspires future research on physical reasoning in settings where ground-truth information is hard to obtain.

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
