# OpenReview forum: "Learning Unsupervised Forward Models from Object Keypoints"
_ICLR.cc/2023/Conference — Submitted to ICLR 2023_

### Official Review · Reviewer_gVjw · 2022-10-24

**Confidence:** 4
**Correctness:** 2
**Technical Novelty And Significance:** 3
**Empirical Novelty And Significance:** 3
**Recommendation:** 5

**Clarity, Quality, Novelty And Reproducibility:**

The paper is generally well-written and easy to follow. However, the novelty of this paper is incremental as supported by the points in the strengths&weakness section. The paper seems to provide sufficient details for reproducing the experiments.

**Strength And Weaknesses:**

[+] The paper proposes to extract object key points as representations for modeling forward dynamics. Such an unsupervised learning setting could be treated as a starting step toward many more research areas.

[+] The experiment section is well-organized with not only an evaluation of the future prediction but also a manipulation task.

[-] One major concern lies in the novelty of this paper. The current method shares the same insight as in previous works like OP3[1] and V-CDN [2] since they both adopt an unsupervised key point/mask detection and then a graph message-passing paradigm for solving the dynamics learning problem. Compared with methods, this paper's novelty seems incremental as they can also handle the generalization to an unseen number of objects. The authors should consider adding proper comparison with these two methods as baselines to make a point on the significance of the current model design.

[-] Meanwhile, the currently selected datasets seem simplistic and fail to illustrate the potential of the model on more complex scenes. Under this setting (2D overhead or top view), the results fail to show the effectiveness of model design (e.g. deterministic vs. probabilistic). The authors should consider settings that better justify the significance of their design, especially when other methods showed more complex scenarios like morphing clothes.

[-] A minor question lies in the design for keypoint detection. Why is the current detector taking as input the pair ($I_0$,$I_t$) but not the more common one ($I_{t-1}$, $I_t$)? I don't see a clear advantage in this design since the model will be required to start prediction from a random initial configuration and always comparing with the initial state will cause trouble in long-horizon planning or prediction (i.e., forgetting in long future predictions).

[1] Veerapaneni et al., "Entity Abstraction in Visual Model-based Reinforcement Learning", CoRL 2019.
[2] Li et al., "Causal Discovery in Physical Systems from Videos", NeurIPS 2020.

**Summary Of The Paper:**

This paper proposes to use key points as internal representations of scenes and model forward dynamics with graph interaction networks. The authors used a multi-object manipulation testbed for evaluating the proposed Keypoint Interaction Network (KINet). They show that KINet can achieve similar results for forward prediction compared with a baseline graph model with ground truth object positions. The authors also show that KINet converges to the goal configuration faster than baselines and generalizes to unseen circumstances where the number of objects, geometries, and background textures are unseen during training.

**Summary Of The Review:**

This paper focuses on learning keypoint-based representations for modeling the forward dynamics of a system. However, the proposed method shares similarity with previous works and the design choices seem to have limited significance as shown in a synthetic dataset setting. Therefore, I don't recommend this paper for acceptance for now. I suggest the authors polish the experiment section and design better experimental settings to show the significance of their method compared with the baselines mentioned in strengths&weakness.

---

> ### Author Response · Authors · 2022-11-19
> **Official Response to Reviewer gVjw**
>
> Thank you for your thoughtful and insightful comments. Please refer to the "Summary of Revisions" comment for a brief summary of changes in the revised manuscript.
>
> > 1. Comparison with more relevant baselines.
>
> Thank you for making this suggestion. We revised Section 2 to discuss more relevant work and highlight the key differences between our approach and existing work. We polished the experiment section and included as a baseline a recent relevant work V-CDN [1] that proposes a keypoint-based forward model. To compare with V-CDN, we generated a more realistic dataset using YCB objects [2] and ran object rearrangement experiments under an unseen number of objects (Figure 5 and Figure 6).
>
> Veerapaneni et al [3] propose an approach based on unsupervised extraction of abstractions in images using an inferred depth to predict the future states of the system. Their model is also used in block stacking action planning using a pick/place action. Our method relies on keypoint extraction for learning a forward model conditioned on sequential pushing actions. To compare with a more recent baseline that also uses keypoints and shares a more similar setup to ours we decided to include V-CDN [1] in this revision but we are open to repurposing [3] as a baseline in our paper in future revisions.
>
> > 2. Performance on more complex scenes.
>
> Thanks for bringing this issue to our attention. We added an additional set of experiments for YCB objects [2]. YCB objects are a set of daily life objects of different properties and complex appearance (shape, texture, and geometry) and are intended for benchmarking in robotic manipulation. We simulated a new dataset using a subset of YCB objects (*YCBObjs* described in Section 4.1 and Section A.1). New experiment results can be found in Figure 5 and Figure 6.
>
> > 3. Minor question.
>
> We decided to use the image pairs $(I_0, I_t)$ over $(I_{t-1}, I_t)$ for training since the dataset is generated by sampling random pushing actions that in some frames do not change the arrangements of the objects. In other words, in our dataset, there are cases where $I_{t-1}$ and $I_t$ remain similar after applying $a_t$. Since the keypoint detection function $f_\textrm{kp}$ relies on moving objects to learn keypoints, it learns more consistent keypoints with a pair of distinct images.
>
> References:
> [1] "Causal discovery in physical systems from videos", Li et al, NeurIPS 2020.
> [2] "The ycb object and model set: Towards common benchmarks for manipulation research.", Calli et al, ICAR 2015.
> [3] "Entity abstraction in visual model-based reinforcement learning." Veerapaneni et al, CoRL 2019.

---

> > ### Comment · Reviewer_gVjw · 2022-12-01
> > **Post-rebuttal response**
> >
> > Thank the authors for their clarifications. I appreciate the additional experiments made on the YCB dataset and the experimental results do favor the proposed KINet over V-CDN. However, my concerns about the significance of the proposed model are not fully eliminated by this additional experiment given that this is the same experiment as the authors originally performed except for the complexity of representing textures. Therefore I have raised my rating to a weak reject hoping that the authors could show the potential of this model in more challenging forward model learning scenarios in future revisions.

---

> > > ### Author Response · Authors · 2022-12-06
> > > **Thank you for your feedback.**
> > >
> > > We sincerely thank you for making valuable suggestions and taking the time to reevaluate our work. We are pleased to hear that the additional experiments further show the advantages of our model.
> > >
> > > Our control experiment setup is a multi-object rearrangement task using sequential pushing actions. This setup is intended for robotic applications and is also used in the baselines we compared with (e.g., Forw [1], ForwInv [1], VisIN [2], V-CDN [3]).
> > >
> > > This experimental setup highlighted the significance of our work in two aspects:
> > > (1) We showed that our unsupervised framework implicitly learns the location and orientation of objects and performs the object rearrangement tasks accurately with no supervision other than image observations.
> > > (2) Importantly, we also showed that our model is able to accurately generalize to an unseen number of objects, object geometries, and background textures. We tested this on datasets that include challenging geometries, complex textures, real-world objects, etc.
> > >
> > > As you suggested, for future work we aim to expand our work to other manipulation settings such as deformable object manipulation. However, for the scope of this paper, we respectfully believe the existing experiments support our claims on the accuracy and generalizability of our model.
> > >
> > > References:
> > > [1] "Learning to poke by poking: Experiential learning of intuitive physics", Agrawal et al, NeurIPS 2016.
> > > [2] "Object-centric forward modeling for model predictive control.", Ye et al, CoRL 2020.
> > > [3] "Causal discovery in physical systems from videos", Li et al, NeurIPS 2020.

---

> ### Author Response · Authors · 2022-11-29
> **Followup with Reviewer gVjw**
>
> Dear Reviewer gVjw,
>
> Thank you again for your thoughtful feedback. Following your suggestions, we have revised our submission and included more evaluations on complex scenarios and real-world objects that highlight the advantage of our approach over baselines. We also posted a global response with summary of all changes [here](https://openreview.net/forum?id=vKEMum01xu&noteId=kSqQGh5523).
>
> We wanted to follow up to see if you have any remaining concerns. Your initial review pointed to one main concern: polishing the experiment section and design better experimental settings to show the significance of their method compared with the baselines. We believe that we have addressed this now, but would appreciate any further feedback.

---

### Official Review · Reviewer_85iU · 2022-10-24

**Confidence:** 3
**Correctness:** 2
**Technical Novelty And Significance:** 2
**Empirical Novelty And Significance:** 3
**Recommendation:** 5

**Clarity, Quality, Novelty And Reproducibility:**

Clarity and quality: I find the paper writing and logic flow clear and easy to follow.

Novelty: The novelty of the proposed model primarily lies in the keypoint abstraction and the interaction network. More justification are expected for why it’s a promising direction.

Reproducibility: Implementation details are not provided in the paper. Code is not included.


**Strength And Weaknesses:**

Strength:
- The idea of combining keypoint abstraction with unsupervised representation learning is novel and the learned forward makes downstream applications like task planning possible.
- The experiments show that the proposed model can achieve high accuracy in forward prediction and the real robot execution task.

Weakness:
- Model design. While combining keypoints and object-centric learning is new, It’s not clear what would be the benefits of choosing a coarser representation as opposed to a denser latent representation. The keypoints abstraction is a bottleneck of the proposed framework, especially in more complex settings or environments. It’s also not obvious how the number of the keypoints $K$ will affect the model's performance as it’s essentially an environment/dataset-dependent hyperparameter. The reconstruction model $f_{rec}$ also depends on $K$. It remains to be seen how the model will perform with more cluttered backgrounds.
- Discussion about more related work. [1] is also a probabilistic object-centric and action-conditioned video prediction model, and interactions between entities are also modeled. [2] Introduces “velocity” and physical interaction and may also be used in modeling action-aware object-centric representations. What are the key differences between the proposed model in terms of the advantages and disadvantages in both representation discovery and forward prediction?
[1] Veerapaneni R, Co-Reyes J D, Chang M, et al. Entity abstraction in visual model-based reinforcement learning[C].
[2] Kossen J, Stelzner K, Hussing M, et al. Structured object-aware physics prediction for video modeling and planning[J].


**Summary Of The Paper:**

This paper proposes an unsupervised framework to learn object-centric representation via keypoint representations. The model abstract the object by keypoints and form a graph representation, where the edges model the relationship between the nodes. An action-conditioned forward model is also learned using contrastive estimation. Experiments are conducted on the rearranging task in MuJoCo as well as a real robot dataset.


**Summary Of The Review:**

My ratings are given based on my evaluation of the strengths and weaknesses above. More justifications are expected in the keypoint-based abstraction and distinction with prior work.

---

> ### Author Response · Authors · 2022-11-19
> **Official Response to Reviewer 85iU**
>
> Thank you for your insightful comments. Please refer to the "Summary of Revisions" comment for a brief summary of changes in the revised manuscript.
>
> > 1. Benefits of keypoints over denser latent representation.
>
> Thank you for pointing this out. The graph in our approach consists of nodes that represent the keypoints. As described in Section 3.2, node embeddings include visual and positional features associated with each keypoint. We compared our approach with Forw and ForwInv baselines (Section 4.2) that use similar supervision as our work (raw visual observation) and learn forward prediction in a dense latent space and showed the effectiveness of our formulation (Table 1, Table 2, and Figure 3). As shown in prior work [1], graphs are a natural representation of multiobject systems and graph representation is effective in modeling such systems due to properties such as order-invariance.
>
> > 2. Effect of the number of keypoints $K$ in complex settings.
>
> Thanks for your thoughtful comment. We simulated a new dataset using a subset of YCB objects [2] (*YCBObjs* in Section 4.1 and Section A.1). YCB objects are a set of daily life objects of different properties and complex appearance (shape, texture, and geometry) and are intended for benchmarking in robotic manipulation.
>
> In complex environments, keypoints can implicitly capture the position and orientation of the objects with no supervision other than visual observation. For instance, consider the keypoints in the qualitative results in Figure 5 where for the power drill object keypoitns are distributed roughly over distinct components of the object without supervision. This allows for an accurate rearrangement of the objects to reach the goal position and orientation. We also included an additional experiment to test for the effect of increasing $K$ in *YCBObjs* dataset (Figure A.3.1).
>
> > 3. Performance on complex backgrounds.
>
> Thanks for making this suggestion. We tested generalization to randomized background complex textures in the object rearrangement experiments. Figure 3b shows the MPC results of zeroshot generalization to randomized textures. Qualitative examples are included in Figure 4 and Figure A.5.2.
>
> > 4. Discussion about relevant work.
>
> Thank you for this suggestion. We revised Section 2 to discuss more relevant work and highlight the key differences between our approach and existing work. We also included as a baseline a recent relevant work V-CDN [3] that proposes a keypoint-based forward model. To compare with V-CDN, we generated a more realistic dataset using YCB objects [2] and ran object rearrangement experiments under an unseen number of objects (Figure 5 and Figure 6).
>
> We discussed the two baselines [4,5] you brought up in the revised relevant work. To summarize, Kossen et al [4] used images to infer explicit state information such as position, velocity, and size in an environment with a fixed number of objects and used graph networks to predict the future appearances of the objects. Unlike our method, this work does not generalize to an unseen number of objects, and it is tested only on 2d geometric shape objects. Veerapaneni et al [5] propose an approach based on unsupervised extraction of abstractions in images using an inferred depth to predict the future states of the system. Their model is also used in block stacking action planning using a pick/place action. Our method relies on keypoint extraction for learning a forward model conditioned on sequential pushing actions. To compare with a more recent baseline that also uses keypoints and shares a more similar setup to ours we decided to include V-CDN [3] in this revision but we are open to repurposing [5] as a baseline in our paper in future revisions.
>
> References:
> [1] "Relational inductive biases, deep learning, and graph networks.", Battaglia et al, CORR 2018.
> [2] "The ycb object and model set: Towards common benchmarks for manipulation research.", Calli et al, ICAR 2015.
> [3] "Causal discovery in physical systems from videos", Li et al, NeurIPS 2020.
> [4] "Structured object-aware physics prediction for video modeling and planning." Kossen et al, ICLR 2020.
> [5] "Entity abstraction in visual model-based reinforcement learning." Veerapaneni et al, CoRL 2019.

---

> > ### Comment · Reviewer_85iU · 2022-12-08
> > **Post-rebuttal**
> >
> > I read the rebuttal and the comments from other reviewers carefully. The concerns from the reviewers are pretty common: the benefits of keypoint presentation, the effect of hyperparameter K, performance on more complex backgrounds, and some missing discussion about related work. I appreciate that the authors address some of them, like including V-CDN as a baseline, ablation on the number of K, and adding discussion on more related work. However, some concerns remain: the effectiveness of the keypoint representation is not fully convincing without more comprehensive comparison plus experiments with prior work; the experiments on K fail to provide more insight on how we should tackle this parameter in the future; the complex background experiments include only textured background, which is still homogeneous. I encourage the authors to improve upon the most significant concerns mentioned above and submit them to later conferences.

---

> > > ### Author Response · Authors · 2022-12-08
> > > **Thank you for your feedback.**
> > >
> > > Thank you for your feedback and for taking the time to read the rebuttal. We are glad to hear that the new datasets and results addressed the common concerns of reviewers. Below we provide a response to your comments:
> > >
> > > > The concerns from the reviewers are pretty common.
> > >
> > > A list of concerns raised by all reviewers is itemized in the [Summary of Revision](https://openreview.net/forum?id=vKEMum01xu&noteId=kSqQGh5523) comment.
> > >
> > > > The effectiveness of the keypoint representation is not fully convincing without more comprehensive comparison plus experiments with prior work;
> > >
> > > The effectiveness of keypoint is demonstrated throughout the paper in two aspects: (1) A keypoint representation allows for learning a forward model without the ground-truth object state information: We showed the prediction accuracy of our model and how it implicitly captures object position and orientation through keypoints (e.g., Figure 5) with no supervision other than rgb image observations. (2) A keypoint representation allows for generalization: In the control experiments, we showed that our model trained on three objects generalizes to an unseen number of objects, unseen background textures, and unseen object geometries.
> > >
> > > We examined our framework on three datasets: a real block dataset, a simulated block dataset with extreme out-of-distribution geometries, and a YCB dataset that includes real-world objects with complex textures that are used for benchmarking in robotic manipulation. We compared our framework with prior work on action-conditional forward modeling from three angles: (1) Forward prediction accuracy (e.g., Table 1). (2) A downstream control task of multiobject rearrangement (e.g., Figure 3 and Table 2) with a setup used in most of the baselines we compared with (e.g., Forw [1], ForwInv [1], VisIN [2], V-CDN [3]). (3) generalization to an unseen number of objects in the object rearrangement task (e.g., Figure 5, Figure 6).
> > >
> > > > ... the experiments on K fail to provide more insight on how we should tackle this parameter in the future;
> > >
> > > Section A.3. provides insight into the effect of the parameter $K$. We examined a wide range of $K=3,\dots, 18$ in the YCB dataset of real-world objects with complex visual features. To summarize, the number of keypoints needs to be at least larger than the number of objects to accurately capture the object geometry and orientation. Keeping all other parameters in the model fixed, increasing the number of keypoints increased the accuracy of the forward prediction for $K=3,\dots, 12$. We observed a slight decrease in the accuracy for $K=15, 18$ which we believe can be mediated by appropriately tunning other hyperparameters such as the keypoint feature map dimension for optimal performance.
> > >
> > >
> > > > ... the complex background experiments include only textured background, which is still homogeneous.
> > >
> > > To show generalization to unseen backgrounds, textures are sampled from standard textures [4] widely used for domain randomization for robotic manipulation. These backgrounds are intended to replicate robotic manipulation environments. As shown in Figure 4 and Figure A.5.2. the backgrounds that we used are: various colors and patterns of wood grain tables, steel tables, bricks, dirt, etc.
> > >
> > >
> > > References:
> > > [1] "Learning to poke by poking: Experiential learning of intuitive physics", Agrawal et al, NeurIPS 2016.
> > > [2] "Object-centric forward modeling for model predictive control.", Ye et al, CoRL 2020.
> > > [3] "Causal discovery in physical systems from videos", Li et al, NeurIPS 2020.
> > > [4] "Robosuite: A modular simulation framework and benchmark for robot learning.", Zhu et al, 2020.

---

> ### Author Response · Authors · 2022-11-29
> **Followup with Reviewer 85iU**
>
> Dear Reviewer 85iU,
>
> Thank you again for your thoughtful feedback. Following your suggestions, we have revised our submission and included more evaluations on complex scenarios and real-world objects that highlight the advantage of our approach over baselines. We also posted a global response with summary of all changes [here](https://openreview.net/forum?id=vKEMum01xu&noteId=kSqQGh5523).
>
> We wanted to follow up to see if you have any remaining concerns. Your initial review pointed to two main concern: model performance under more complex scenarios, and more discussion and comparison to existing methods. We believe that we have addressed these now, but would appreciate any further feedback.

---

### Official Review · Reviewer_2bb6 · 2022-11-04

**Confidence:** 4
**Correctness:** 3
**Technical Novelty And Significance:** 2
**Empirical Novelty And Significance:** 2
**Recommendation:** 5

**Clarity, Quality, Novelty And Reproducibility:**

Clarity:
1. Overall the method and experiments are clear.

Novelty:
1. This method looks very similar to Ye(2020). The main difference in the object representation learning part is that in Ye(2020), they require the ground-truth object locations. The main difference in the forward model part is the probabilistic graph representation. I am not entirely convinced that the novelty is crucial. Presenting more comprehensive baselines and experiments may help.

**Strength And Weaknesses:**

Strengths:
1. This paper presents a method to learn unsupervised forward models that can be generalized to different number of objects.
2. The method is straightforward and easy to understand.

Weaknesses:
1. In the related work section, authors mention that Kipf et al.(2019) is very relevant to your method. Kipf(2019) is also unsupervised and doesn't require ground-truth object locations. Could you include this method as part of the baseline as well?
2. Another natural baseline would be Minderer et al.(2019), which also learns a dynamics model base on unsupervised keypoint representations.
3. One of the main contribution is the probabilistic graph representation enabled by the learned adjacency matrix. The insight behind this design is that keypoints that are closer in the space could provide redundant information. However in the visualization of the adjacency matrix(supplement A.2.1), I can't reach to a conclusion as in what's being learned in these graphs. Can you provide more explanation and experiments to prove that this design is essential to the method(other than KINet-deterministic)?
4. I am curious how would the method perform with more keypoints. If there are similar number of keypoints and objects in the scene, each object is only represented by one or two keypoints. Since each object is only represented by their keypoints, is one or two keypoints enough to capture the pose and orientation information?


**Summary Of The Paper:**

This paper presents a method that learns dynamics models from object-centric representations in manipulation settings. This method is unsupervised and can generalize to different numbers of objects, unseen object shapes and unseen backgrounds. They demonstrate the effectiveness of this forward models by applying it to downstream robotics control tasks with a graph MPC algorithm.

**Summary Of The Review:**

This paper presents a new method on unsupervised learning of forward models based on probabilistic graph representations. I am not convinced that the novelties presented in this method are crucial. Presenting more comprehensive comparison to existing methods and other suggested baselines may help.

---

> ### Author Response · Authors · 2022-11-19
> **Official Response to Reviewer 2bb6 (2/2)**
>
> References:
> [1] "Causal discovery in physical systems from videos", Li et al, NeurIPS 2020.
> [2] "The ycb object and model set: Towards common benchmarks for manipulation research.", Calli et al, ICAR 2015.
> [3] "Unsupervised learning of object structure and dynamics from videos." Minderer et al, NeurIPS 2019.
> [4] "Contrastive learning of structured world models." Kipf et al, ICLR 2019.
> [5] "Object-centric forward modeling for model predictive control.", Ye et al, CoRL 2020.

---

> ### Author Response · Authors · 2022-11-19
> **Official Response to Reviewer 2bb6 (1/2)**
>
> Thank you for your insightful comments. Please refer to the "Summary of Revisions" comment for a brief summary of changes in the revised manuscript.
>
> > 1. Including more relevant baselines.
>
> Thank you for making this suggestion. We revised Section 2 to discuss more relevant work and highlight the key differences between our approach and existing work. We also included as a baseline a recent relevant work V-CDN [1] that proposes a keypoint-based forward model. To compare with V-CDN, we generated a more realistic dataset using YCB objects [2] and ran object rearrangement experiments under an unseen number of objects (Figure 5 and Figure 6).
>
> We discussed the two interesting methods [3,4] you brought up in the revised relevant work. To summarize, for action planning we looked for baselines that are formulated as an action-conditioned model. Minderer et al [3] propose a keypoint-based video prediction framework that predicts feasible future frames given a history of the previous frames. Without action as an input, this model cannot be applied to our setting in which each frame is recorded based on an independent randomized action. Kipf et al [4] proposed unsupervised extraction of abstractions in images to predict future appearances using contrastive loss and examined their method on multibody environments with deterministic structures and minimal visual features such as 2d shapes. To compare with a more recent baseline that also uses keypoints and shares a more similar setup to ours we decided to include V-CDN [1] in this revision but we are open to including [4] as a baseline in future revisions.
>
> > 2. Novelty.
>
> We would like to address your concern about the novelty of our work compared to Ye et al [5] and highlight the advantages of our proposed approach. In Ye et al, the scene with a fixed number of objects is represented as a graph where each node embedding includes the ground-truth object location and visual features from a bounding box around the ground-truth location (replicated in VisIN baseline). As mentioned in the conclusion of their work, this work has two main limitations: (1) explicit supervision on the location of the objects is not always feasible (2) building a graph where each node represents an object makes the learned model dependent on the number of objects and does not generalize to an unseen number of objects.
>
> Our work proposes a framework for unsupervised forward prediction using no supervision other than raw visual observations. Forward prediction in the keypoint space relaxes the assumption of having access to ground-truth object states. Keypoints implicitly capture the position and orientation of objects (Figure 5). Importantly, our formulation does not depend on the number of objects, and we demonstrate its generalizability to unseen circumstances such as an unseen number of objects using YCB objects and block objects.
>
> > 3. Justification for a probabilistic graph representation.
>
> Thanks for this thoughtful question. As you mentioned, we showed the effectiveness of a probabilistic graph representation over a deterministic fully connected graph (KINet-deterministic) in prediction accuracy (Table 1). We also showed a probabilistic representation better generalizes to an unseen number of objects (Table 2). Although we do not explicitly explore the interpretability in this work, Figure A.2.1 has been included to visualize the learned node embeddings which show keypoints that were assigned to a specific object are distinctively aligned in a 2d projection of the feature space.
>
> > 4. Capturing pose and orientation information with more keypoints.
>
> Thanks for pointing this out. We included a new dataset with more complex objects and ran object rearrangement experiments to test if with our method keypoints are able to capture pose and orientation. The new dataset is simulated using a subset of YCB objects [2] (*YCBObjs* in Section 4.1 and Section A.1). YCB objects are a set of daily life objects of different properties and complex appearance (shape, texture, and geometry) and are intended for benchmarking in robotic manipulation.
>
> For instance, consider the keypoints in the qualitative results in Figure 5 where for the power drill object keypoints are distributed roughly over distinct components of the object with no supervision which implicitly captures the object orientation as well as its position. This allows for an accurate rearrangement of the objects to reach the goal configuration. We also included an additional experiment to test for the effect of increasing $K$ in *YCBObjs* dataset (Figure A.3.1).

---

> > ### Comment · Reviewer_2bb6 · 2022-12-09
> > **Post-rebuttal response**
> >
> > Thank you for your additional experiments and discussion about additional related work. One minor mistake: the authors discuss [4] in the related work section about forward models. However in [4], visual features are represented as object segmentation masks, which are jointly learned with the forward transition models without explicit supervision. This undermines the claim that part of the novelty comes from inferring the object locations/orientations without supervision.
> > Given the additional experiments, my concerns about the novelty of the method still remain. Comparing the results from KINet-deterministic, KINet and VisIn, it seems that the benefit from having a probabilistic graph model is limited, and there seems to be a larger discrepency by replacing GT object states with learned keypoints. These undermine the significance of the two design decisions this method makes. Therefore I remain my original score.

---

> > > ### Author Response · Authors · 2022-12-09
> > > **Thank you for your feedback.**
> > >
> > > Thank you for your feedback and for taking the time to read our rebuttal. Below we provide a response to your comments:
> > >
> > > >  One minor mistake: the authors discuss [4] in the related work section about forward models.
> > >
> > > We discuss [4] in a dedicated subsection of "Unsupervised Forward Models" under Section 2. We summarized this work as "an unsupervised forward model that extracts object-centric abstractions in multibody environments with deterministic structures and minimal visual features such as 2d shapes". We also cited [4] in the methods that follow the Interaction Network under the "Forward Models" subsection but we can see how this can be confusing and will remove the citation from this subsection.
> > >
> > > > .. in [4], visual features are represented as object segmentation masks, which are jointly learned with the forward transition models without explicit supervision. This undermines the claim that part of the novelty comes from inferring the object locations/orientations without supervision.
> > >
> > > First, we emphasize that objects in [4] are simple geometries with fixed minimal visual features. The objects are also only translated in 2D with no rotation. To highlight the difference, we encourage the reviewer to compare the complexity of the objects in [4] with our experimental setting (please compare the 2d circles and squares in Fig. 2 of [[4]](https://arxiv.org/pdf/1911.12247.pdf) with YCB objects in Fig. 5 of our paper).
> > >
> > > It is a stretch to conclude that object-centric masks learned in [4] are "segmentation" masks that represent object location and orientation. We think this is also not the claim of the authors of [4]. An example of the inferred masks is provided in Fig. 3-a of [4]. Even in the "2D Shape" environment with fixed objects, it is obvious that the predicted masks only capture the object's location and not the object's geometry and orientation.
> > >
> > > We respectfully cannot see how the framework of [4] "undermines" our claim that our keypoint-based method implicitly captures the object location and orientation. As shown in Fig. 5, we demonstrate how keypoints bind to distinct parts of each object and collectively represent their location and orientation in an unsupervised manner.
> > >
> > > > Comparing the results from KINet-deterministic, KINet and VisIn, it seems that the benefit from having a probabilistic graph model is limited.
> > >
> > > We respectfully think the reviewer is not including Table 2 in their assessment. Results in Table 2 show that a probabilistic graph better generalizes to an unseen number of objects (please compare KINet with KINet-deterministic in Table 2). As discussed in Section 5.4., with a probabilistic graph representation, our model achieves significantly better generalization to an unseen number of objects and unseen geometries. This performance gap is more evident when generalizing to unseen geometries. We also provided this information in our initial response.
> > >
> > > > ... and there seems to be a larger discrepency by replacing GT object states with learned keypoints. These undermine the significance of the two design decisions this method makes.
> > >
> > > We showed the significance of our model design from two major aspects: (1) Not relying on the ground truth object states: it is not always feasible to assume access to such ground-truth information, especially in robotic manipulation. (2) Generalization to unseen circumstances: we demonstrated how our work stands out among the existing baselines on action-conditional forward modeling from a generalization aspect which again is crucial for robotic manipulation.

---

> ### Author Response · Authors · 2022-11-29
> **Followup with Reviewer 2bb6**
>
> Dear Reviewer 2bb6,
>
> Thank you again for your thoughtful feedback. Following your suggestions, we have revised our submission and included more evaluations on complex scenarios and real-world objects that highlight the advantage of our approach over baselines. We also posted a global response with summary of all changes [here](https://openreview.net/forum?id=vKEMum01xu&noteId=kSqQGh5523).
>
> We wanted to follow up to see if you have any remaining concerns. Your initial review pointed to one main concern: more discussion and comparison to existing methods to highlight novelties. We believe that we have addressed this now, but would appreciate any further feedback.

---

### Official Review · Reviewer_E8Ms · 2022-11-04

**Confidence:** 4
**Correctness:** 3
**Technical Novelty And Significance:** 3
**Empirical Novelty And Significance:** 2
**Recommendation:** 8

**Clarity, Quality, Novelty And Reproducibility:**

The paper is well written and easy to follow. Reproducibility seems high (although would be even higher if the source code and simulation data is released - but I didn’t see that mentioned anywhere in the paper).

Experimental results - while limiting (see above) - seem well carried out and of high quality (e.g. multiple seeds with sigma values given).

Originality of the work: There is some novelty in the graph formulation as far as I know. Unsupervised keypoint detection for use in dynamics modeling has certainly been done a few times at this point (e.g. the cited paper Manuelli et al., Keypoints into the future). However I think there’s enough novelty here to justify the submission.

**Strength And Weaknesses:**

Strengths:

- Not needing ground truth keypoints is an obvious win. It should make this method usable in multiple setups.
- The keypoint extraction and graph association are object agnostic, meaning that the method should generalize to unseen objects and scenes. That is, the keypoints have no specific semantic encoding that would cause generalization to fail to other classes of unseen objects.

Weaknesses:

- The fixed “K” in the keypoint detector seems particularly problematic. If K is overcomplete (i.e. just some huge value) for describing the intrinsic degrees of freedom of an object, you could argue that perhaps a fixed K is OK, but generally, for articulated or deformable objects, you might not know a priori the “K” value that is needed to fully recover the pose of an object. Another argument to make is that the k-dimensional feature map is just yet another hidden layer of a larger network (that happens to include the graph representations) - in which case it’s probably a stretch claiming these are semantic keypoints at all.
- Related to above, the authors claim that the K representation is a bottleneck, however how much of a bottleneck seems to be very much dependent on the value of “K”.
- Addition of the contrastive loss seems like a tacked-on solution and the authors need to do a better job describing what problem it solves (or what additional regularization it adds). The ablation where it is removed seems fine, but this alone is not enough to justify its use.
- The class of objects the proposed method is evaluated on seems particularly limiting. The real-world results are all roughly rectangular shapes on a consistent background with minimal occlusion (there’s some slightly more complicated shapes in the sim experiments, but even these are simple from an appearance perspective). I would like to see how this method performs on more complex scenes with background clutter and even on articulated or soft-bodied objects. Why is this not unreasonable or out of scope for this paper?: because the promise of this unsupervised method is that it works on any objects without needing semantic labels, however these experimental results fall short of showing that IMO. By contrast, for THIS particular application, you could very likely do simple heuristics to get a set of keypoints that would probably work just as well (e.g. color centroids, etc).
- Nit: The Fig 2 real-robot data does not feel like “robot data” to me (I work primarily in robotics). There’s no robotic embodiment visible in the frame, no active perception, etc. For instance a human could have moved the objects between frames. This is a perfectly fine computer vision dataset, but I think it’s a stretch to claim it’s somehow robot data. 5.2 presumably uses a robot (with MPC on your forward model) to actually move the objects, so why isn’t this visible in the frames?

**Summary Of The Paper:**

The authors present Keypoint Interaction Network (KINet) - an unsupervised learning method to associate keypoints with objects and uses a forward motion model to estimate future keypoint states. As an unsupervised method, the authors make no assumptions about access to ground truth semantic keypoint locations, but rather just use image observations and actions.

**Summary Of The Review:**

Overall the paper seems novel and is well written. It is likely to be of interest to the community, however I do find the experimental results to be somewhat underwhelming (hence the final score). I’d like to see more real-world examples with more complex object and scene appearances.

---

> ### Author Response · Authors · 2022-11-19
> **Official Response to Reviewer E8Ms**
>
> Thank you for your insightful comments. Please refer to the "Summary of Revisions" comment for a brief summary of changes in the revised manuscript.
>
> > 1. The choice of hyperparameter $K$.
>
> Thank you for pointing this out. We agree that the choice of $K$ is largely environment dependent. The keypoint bottleneck $f_{\mathrm{kp}}$ ensures that the visual features of the moving elements in the image (between $I_0$ and $I_t$) are extracted over $K$ feature maps that can be used to reconstruct $f_{\mathrm{rec}}$ the visual observation. By incorporating the reconstruction loss, the keypoints track specific visual features in objects and therefore are semantic.
>
> We have included new results to highlight this. For instance, consider the keypoints in the qualitative results in Figure 5 where for the power drill object keypoitns are distributed roughly over distinct components of the object with no supervision. This allows for an accurate rearrangement of the objects to reach the goal position and orientation. We also included an additional experiment to test for the effect of increasing $K$ in *YCBObjs* dataset (Figure A.3.1).
>
> > 2. Justification for contrastive loss.
>
> Thanks for this observation. We updated Section 3.5 to motivate the inclusion of contrastive loss and describe its effect in addition to the ablation result (Table 1). To summarize, learning good graph representation is a challenge, especially in an unsupervised framework. Intuitively, contrastive loss aligns the graph representation of similar object configurations while pushing away the graph representation of the dissimilar object configurations in the embedding space which enhances the learned graph representations.
>
> > 3. Performance on more complex objects.
>
> Thanks for making this suggestion. We added an additional set of experiments for YCB objects [1]. YCB objects are a set of daily life objects of different properties and complex appearance (shape, texture, and geometry) and are intended for benchmarking in robotic manipulation. We simulated a new dataset using a subset of YCB objects (*YCBObjs* described in Section 4.1 and Section A.1).
>
> We ran extensive additional experiments to test our approach on *YCBObjs* dataset. After training on a subset of 3 YCB objects, we tested for generalization to an unseen number of YCB objects (1,2,4,5) in an object rearrangement control task. We found that our framework generalizes to an unseen number of YCB objects and accurately rearranges them to the goal configuration (see Figure 5 for qualitative and Figure 6 for quantitative MPC results). In this paper, we are mainly focusing on rigid objects to highlight the effectiveness of an unsupervised forward model for generalization to unseen scenarios. We plan to extend our method for articulated and deformable objects in future work.
>
> > 4. The presence of a robot.
>
> Thank you for this thoughtful question. In our simulated experiments, the end-effector is a rigid black cuboid (shown in Figure A.1.2) that is removed before taking images (mentioned in section A.1). Also, in the real-robot dataset [2], the Sawyer robot arm is moved out of the workspace after the pushing action is performed. Since we are relying solely on single-view visual observations to learn an unsupervised forward model, the presence of a robot end-effector introduces further complexities to the problem we are solving such as obstructing the scene. Therefore, in this work, we made an assumption that after applying the action the end-effector can be lifted from the scene before a visual observation is made. As a future work, we plan to relax this assumption by conditioning the state of the robot manipulator (e.g., joints, gripper) to our model so that it would learn to distinguish object keypoints from robot keypoints.
>
> References:
> [1] "The ycb object and model set: Towards common benchmarks for manipulation research.", Calli et al, ICAR 2015.
> [2] "Object-centric forward modeling for model predictive control.", Ye et al, CoRL 2020.

---

> > ### Comment · Reviewer_E8Ms · 2022-12-06
> > **Increasing score.**
> >
> > The reviewers have addressed many of my concerns and I'm going to increase my score accordingly.

---

> > > ### Author Response · Authors · 2022-12-06
> > > **Thank you for your feedback.**
> > >
> > > We are very pleased to hear that our rebuttal addressed your concerns. We sincerely thank you for making valuable suggestions and taking the time to reevaluate our work.

---

> ### Author Response · Authors · 2022-11-29
> **Followup with Reviewer E8Ms**
>
> Dear Reviewer E8Ms,
>
> Thank you again for your thoughtful feedback. Following your suggestions, we have revised our submission and included more evaluations on complex scenarios and real-world objects that highlight the advantage of our approach over baselines. We also posted a global response with summary of all changes [here](https://openreview.net/forum?id=vKEMum01xu&noteId=kSqQGh5523).
>
> We wanted to follow up to see if you have any remaining concerns. Your initial review pointed to one main concern: more real-world examples with more complex object and scene appearances. We believe that we have addressed this now, but would appreciate any further feedback.

---

### Author Response · Authors · 2022-11-19
**Summary of Revisions**

We thank all reviewers for their time and their insightful feedback and suggestions.

We are pleased that the reviewers found our work well written and straightforward to follow [$\textcolor{purple}{\textrm{E8Ms}}$, $\textcolor{navy}{\textrm{2bb6}}$, $\textcolor{teal}{\textrm{85iU}}$, $\textcolor{green}{\textrm{gVjw}}$] with clear and well-carried experiments [$\textcolor{purple}{\textrm{E8Ms}}$, $\textcolor{navy}{\textrm{2bb6}}$, $\textcolor{green}{\textrm{gVjw}}$] and acknowledged that our unsupervised forward prediction approach is an obvious win [$\textcolor{purple}{\textrm{E8Ms}}$] with high generalizability [$\textcolor{purple}{\textrm{E8Ms}}$, $\textcolor{navy}{\textrm{2bb6}}$, $\textcolor{green}{\textrm{gVjw}}$] that applies to various setups [$\textcolor{purple}{\textrm{E8Ms}}$, $\textcolor{teal}{\textrm{85iU}}$] and can be a starting point in many research areas [$\textcolor{green}{\textrm{gVjw}}$].

We believe that through your assistance the paper has improved. Below we summarize the updates in our revision following the suggestions of reviewers:

1. **[Section 2. Related work]** [$\textcolor{navy}{\textrm{2bb6}}$, $\textcolor{teal}{\textrm{85iU}}$, $\textcolor{green}{\textrm{gVjw}}$] More relevant approaches are added in the unsupervised forward models subsection and key differences with existing work are highlighted.
2. **[Section 3.5. Loss]** [$\textcolor{purple}{\textrm{E8Ms}}$]  Motivation for adding contrastive loss term is clarified.
3. **[Section 4.1. Dataset]** [$\textcolor{purple}{\textrm{E8Ms}}$, $\textcolor{navy}{\textrm{2bb6}}$, $\textcolor{teal}{\textrm{85iU}}$, $\textcolor{green}{\textrm{gVjw}}$] New dataset is added using YCB object dataset to generate more realistic scenarios using objects with complex properties.
4. **[Section 4.2. Baselines]** [$\textcolor{navy}{\textrm{2bb6}}$, $\textcolor{teal}{\textrm{85iU}}$, $\textcolor{green}{\textrm{gVjw}}$] New baseline is included to compare our approach with a recent unsupervised keypoint-based forward model.
5. **[Section 5.3. Results]** [$\textcolor{purple}{\textrm{E8Ms}}$, $\textcolor{navy}{\textrm{2bb6}}$, $\textcolor{teal}{\textrm{85iU}}$, $\textcolor{green}{\textrm{gVjw}}$] Thorough experiments are added to examine the performance of our approach on the challenging YCB objects. Object rearrangement control task and generalization to an unseen number of objects are compared with the new baseline.
6. **[Appendix A.1. Dataset]** Details of new YCB objects dataset are included.
7. **[Appendix A.3. Additional Results]** [$\textcolor{purple}{\textrm{E8Ms}}$, $\textcolor{teal}{\textrm{85iU}}$] The effect of varying the number of keypoints on the performance of our model is examined.
8. **[Appendix A.5. Additional Results]** Qualitative results of zeroshot generalization to randomized background textures are moved to the appendix to make space for new results.

---

### Decision · Program_Chairs · 2023-01-20

**Decision:**

Reject

**Justification For Why Not Higher Score:**

Incremental technical contributions and insufficient evidence to validate them.

**Justification For Why Not Lower Score:**

N/A

**Metareview: Summary, Strengths And Weaknesses:**

This paper jointly trains a keypoint representation and a forward model on it, from observing videos of interacting objects. The keypoint representation loss is based on Kulkarni et al 2019 (Tranporter), and the forward model is a graph-based dynamics model along the lines of interaction networks (Battaglia et al 2016), but with probabilistic edges.

Strengths:
- During the rebuttal stage, the authors added a new dataset YCBObjs, that contains quite complex shapes, and on which the method performs quite well (Fig 5). Additionally, the improvements during the rebuttal stage included a new and rather recent baseline, that make for a fairly thorough evaluation.

Weaknesses:
- The technical contributions are quite incremental, well approximated as a straightforward combination of Transporter and Interaction Nets.
- The one key difference from this above approximation, the probabilistic edges in the graph, doesn't contribute to measurable performance improvements, as seen in Tab 1 for example (KINet vs KINet - deterministic).
- The idea that this method can work even with novel objects is a bit odd. This is not a claim made by prior work on keypoint detection, such as Transporters, which this builds on. It would be appropriate to be more precise with this statement. What kind of generalization is being claimed here, and is this improving on the generalization that Transporters already provide for keypoint detection? If so, how is this improvement achieved?

Another note to the authors (did not affect my recommendation):
- The right reference for model-predictive control is not Finn and Levine 2017. See e.g. Garcia, Carlos E., David M. Prett, and Manfred Morari. "Model predictive control: Theory and practice—A survey." Automatica 25.3 (1989): 335-348.

- In Sec 5.2, Note that planning horizons of T=40 for these types of tasks are rather long, particularly for pixel-based models such as "Forward" and "Forward-Inverse" among the baselines.